# Parasites of Free-Ranging and Captive American Primates: A Systematic Review

**DOI:** 10.3390/microorganisms9122546

**Published:** 2021-12-09

**Authors:** Silvia Rondón, Serena Cavallero, Erika Renzi, Andrés Link, Camila González, Stefano D’Amelio

**Affiliations:** 1Department of Public Health and Infectious Diseases, Sapienza University of Rome, Piazzale Aldo Moro 5, 00185 Rome, Italy; serena.cavallero@uniroma1.it (S.C.); erika.renzi@uniroma1.it (E.R.); stefano.damelio@uniroma1.it (S.D.); 2Laboratorio de Ecología de Bosques Tropicales y Primatología, Departamento de Ciencias Biológicas, Universidad de Los Andes, Cra. 1 N° 18a-12, Bogotá 111711, Colombia; a.link74@uniandes.edu.co; 3Centro de Investigaciones en Microbiología y Parasitología Tropical, CIMPAT, Departamento de Ciencias Biológicas, Universidad de los Andes, Cra. 1 N° 18a-12, Bogotá 111711, Colombia; c.gonzalez2592@uniandes.edu.co

**Keywords:** American non-human primates, parasites, zoonosis, diagnostic methods

## Abstract

The diversity, spread, and evolution of parasites in non-human primates (NHPs) is a relevant issue for human public health as well as for NHPs conservation. Although previous reviews have recorded information on parasites in NHPs (Platyrrhines) in the Americas, the increasing number of recent studies has made these inventories far from complete. Here, we summarize information about parasites recently reported in Platyrrhines, attempting to build on earlier reviews and identify information gaps. A systematic literature search was conducted in PubMed, ISI Web of Science, and Latin American and Caribbean Health Sciences Literature (LILACS), and following the Preferred Reporting Items for Systematic Reviews and Meta-analyses (PRISMA) guidelines. Ninety-three studies were included after the screening process. Records for 20 genera of NHPs, including 90 species were found. Most of the studies were conducted on captive individuals (54.1%), and morphological approaches were the most used for parasite identification. The most commonly collected biological samples were blood and stool, and Protozoa was the most frequent parasite group found. There is still scarce (if any) information on the parasites associated to several Platyrrhine species, especially for free-ranging populations. The use of molecular identification methods can provide important contributions to the field of NHPs parasitology in the near future. Finally, the identification of parasites in NHPs populations will continue to provide relevant information in the context of pervasive habitat loss and fragmentation that should influence both human public health and wildlife conservation strategies.

## 1. Introduction

Public health, animal welfare, and pathogen transfer to and from wild populations are among the current primary issues of concern in the framework of the One-Health concept. Such aspects are even more relevant in areas of the world such as South America, where biodiversity is declining at high rates and the rate of deforestation is growing. There is compelling evidence on how habitat loss and fragmentation may favor contact between humans and other animals, representing a potential threat for both [1]. In this scenario, non-human primates (NHPs) are of particular interest because of their close phylogenetic relationship with humans and their known role as reservoirs of zoonotic agents [2]. 

So far, six major groups of organisms have been found infecting NHPs: viruses, bacteria, fungi, protozoa, helminths, and arthropods [3]. For a series of multiple issues including behavioral ecology, public health, and NHPs conservation, it is important to understand the diversity, spread, and evolution of parasites in wild NHPs [4]. Despite this, the inventory of parasites infecting American NHPs is far from complete, highlighting the need for interdisciplinary studies aiming to determine and treat NHPs parasites [5].

Among mammals, the Order Primates includes a high number of species classified as threatened according to the International Union for Conservation of Nature (IUCN), and specifically in Latin America 9% of NHPs species are considered as critically endangered, 12.4% as endangered, and 20.3% as vulnerable according to the Red List [6]. Habitat loss and forest fragmentation are some of the main threats to NHPs species [7], while livestock and ranching are secondary threats affecting 59% of NHPs species in the Neotropics [8].

The implementation of effective measures to reverse anthropogenic pressures against NHPs populations has been strongly encouraged to avoid the imminent loss of NHPs taxa, due to factors such as fragmented landscapes, habitat loss, and degradation, as well as human and domestic animal-borne diseases [8]. As infectious diseases negatively impact NHPs populations, it becomes necessary to watch over all possible introductions of disease, and to better investigate the correlation between diversity and disease exposure risk in humans and wildlife [9]. According to the World Organisation for Animal Health (OIE), the probability of carrying zoonotic pathogens is related to the taxonomic position (increasing from lemurs and tarsiers to marmosets, tamarins and other American monkeys, and finally African and Asian monkeys and apes) and to the region of origin of the species of concern [10]. Likewise, the risk of zoonotic infections involving NHPs is of public health concern, as the expanding human–domestic animal–wildlife interface provides multiple opportunities for the agents of disease to shift hosts. Additionally, fragmentation and habitat loss interrupt natural processes involving parasites and hosts [11].

In this context, molecular epidemiology and diagnosis of parasitic zoonoses in NHPs play a very important role in the understanding of parasite ecology and the assessment of their zoonotic potential. Previous reviews have gathered information of parasites in Platyrrhines [5,12] or have been focused on the fragmentation of the fauna living in the American tropics [13,14]. Given the large number of recent studies, it is worth updating and compiling NHPs parasitological information along with information related to threatening factors, creating a useful tool that may serve as the basis to better direct future research projects, support decision making and fill information gaps.

This systematic review summarizes information about parasites (protozoans, helminths and ectoparasites) recently reported in American NHPs, building on previous reviews and attempting to identify information gaps regarding the parasitic pathogens circulating in a major concern group of hosts and in a critical biodiverse region.

## 2. Materials and Methods

We carried out a systematic review following the Preferred Reporting Items for Systematic Reviews and Meta-Analyses (PRISMA) to summarize information about parasites infecting American NHPs. The review protocol of this systematic review was not recorded into the International prospective register of systematics reviews (PROSPERO) (Appendix A). We performed an independent search for each Platyrrhine genus, using the terms “parasite” and NHPs genus (e.g., parasite AND *Cebus*). The search was conducted in ISI Web of Knowledge and PubMed, including studies from June 2017 to 11 February 2021, thus collecting all the information published after the time frame used in the last available review regarding the subject [5]. Additionally, information from Latin American and Caribbean Health Sciences Literature (LILACS) was incorporated into the database, using the same search terms, until February 11th, 2021. In this way we collected information from a Latin American specific search engine, building on the review made by Solórzano-García and Pérez-Ponce de León [5].

We included studies performed in wild and captive Platyrrhines which reported parasite occurrence, while studies under laboratory conditions or the ones focused on fungi, bacteria, and viruses were not included. We used articles in English, Portuguese, or Spanish.

Two reviewers screened the records independently. In case of a disagreement that was not consensually solved, a third reviewer arbitrated the decision process. For data extraction, we used a standardized form that included the following features: host family, host genera, host species, collected sample (stool, blood, tissue, ectoparasite), parasite detection method (PCR, microscopy, etc.), parasite family, parasite genus, parasite species, parasite prevalence (%), parasite group (protozoa, cestoda, nematoda, trematoda, phthiraptera, acariformes, ixodida, diptera, pentastomida, siphonaptera), endoparasite/ectoparasite, forest fragmentation evaluated (yes/no), country, state and habitat (sylvatic/captivity).

To organize the collected data, we considered specific taxonomy classifications: for *Lagothrix*, *Saguinus*, and *Callicebus*, we followed the classification proposed by Di Fiore et al. [15], Buckner et al. [16], and Byrne et al. [17], respectively. For all other NHPs genera, we followed the taxonomy of the “Handbook of the mammals of the world” [18]. Parasite taxonomy was included following the classification stated by the National Center for Biotechnology Information (NCBI). 

## 3. Results

The literature review retrieved 720 searches: 444 from ISI Web of Knowledge, 214 from PubMed, and 62 from LILACS. Overall, we obtained 93 novel publications (Figure 1) after eliminating duplicates, studies under laboratory conditions, studies already included in the review made by Solórzano-García & Pérez-Ponce de León [5].

Overall, the studies included in this review account for 20 Platyrrhine genera, including 90 species. The genus with most records was *Alouatta* (*n* = 51), while genera with the least records were *Callimico* (*n* = 1) and *Cebuella* (*n* = 1) (Table 1). According to the parasite group, protozoa were overall the most reported along NHPs genera (Table 1). It was found that 54.1% studies were conducted on captive NHPs and 45.9% on free-ranging animals, while the source of biological sample and diagnostic method mostly used were blood and morphology, respectively (Table 2). 

A list of parasites per NHPs species is shown (Table 3), as well as a list of parasite-host (Appendix A). When considering the geographical distribution of the records, Brazil was the country with most of them, including information for 19 NHPs genera. There were no publications for Belize, Bolivia, Guatemala, Guyana, Honduras, El Salvador, Suriname, and Venezuela (Figure 2). Additionally, there were publications regarding captive Neotropical NHPs in Europe and Asia: *Aotus*, *Callimico*, and *Cebuella* in Switzerland (1 record each), *Cebus* in France (1 record), *Saguinus* in Italy (1 record), *Ateles*, *Saimiri*, and *Sapajus* in China (1, 2, and 1 records, respectively), *Callithrix* in Korea (1 record), and *Saimiri* in Japan and South Korea (1 record for each country). 

*Alouatta* was the NHPs genus with most records (*n* = 51) followed by *Callithrix* (*n* = 22), while for all other genera less than 20 records were retrieved (Table 1). Considering the number of species, the genera with more species are *Plecturocebus* (*n* = 23), *Saguinus* (*n* = 15), *Cebus* (*n* = 14), *Alouatta* (*n* = 12), and *Aotus* (*n* = 11). Likewise, *Ateles*, *Brachyteles*, *Callimico*, *Callithrix*, *Cebuella*, and *Sapajus* had records for 100% of the species, and 80% of the species for *Lagothrix*, *Pithecia*, and *Saguinus*. Brazil was the country with most parasitological records (*n* = 163), being also the country with highest recorded occurrence of NHPs. Other NHPs rich-countries, such as Peru and Colombia, were the second and third countries with more records (20 and 13, respectively).

## 4. Discussion

The most recent list (2018–2020) of the World’s 25 Most Endangered NHPs Species includes six Platyrrhines: *Ateles geoffroyi*, *Cebus aequatorialis*, *Saguinus bicolor*, *Plecturocebus olallae*, *Alouatta guariba*, and *Callithrix aurita* [112]. After the systematic review process, there were retrieved publications with parasitological data for *A. geoffroyi* (*n* = 4), *S. bicolor* (*n* = 4), *A. guariba* (*n* = 11), and *C. aurita* (*n* = 1), while there were no articles mentioning *C. aequatorialis* and *P. olallae*. Additionally, there were no records for *Plecturocebus caquetensis*, *P. olallae*, *Leontopithecus caissara*, and *Callicebus barbarabrownae*, listed in the IUCN Red List as Critically Endangered [6], neither for *Cebus malitiosus*, *Saimiri vanzolinii*, *Callicebus coimbrai*, *Alouatta ululata*, or *Cebus cesarae*, listed as Endangered [6]. The amount of information is probably biased by the availability of different species in captivity, a condition that strongly facilitate parasitological investigations. It can be speculated that the lack of information for endangered species could be related to their scarcity in captive conditions. Although other kinds of studies (e.g., behavioral, genetic) may have been carried out for those species during the time range considered in this study, it must be highlighted that parasitological studies are also very important, representing a useful insight for monitoring the health status of NHPs in contexts of human–NHPs interfaces, as human-induced forest loss increase the exposition of NHPs to human and domesticated animal pathogens [8]. Additionally, even if non-lethal parasite infections are common in wild NHPs, parasite infections could cause sickness behaviors that may be adaptative in the short-term but have longer-term fitness consequences [113]. Note that for some Critically Endangered and Endangered NHPs species which had no parasitological studies until 2017, data have been recorded between 2017 and 2021, as is the case of *Cebus kaapori*, *Sapajus flavius*, and *Ateles marginatus*. Moreover, even if there are reports for specific NHPs species, the observation is limited to a specific area implying that not all the geographic range of the species has been covered.

Overall, just over half of the studies were conducted on captive NHPs (54.1%), however, for the genera *Alouatta*, *Cacajao*, *Callithrix*, and *Leontocebus* there were more records on free-ranging NHPs. Studies in both free-ranging and captive NHPs are important, for instance, in the design of conservation strategies, reintroduction programs, and NHPs acquisition for research laboratories or zoos. Determining the composition of parasite communities in captive NHPs allows the identification of parasites of concern regarding the introduction of novel parasites to potentially susceptible wildlife populations during reintroduction programs, and also lead to a better understand parasite ecology, for instance, it has been observed that vector-borne parasites are more likely found in free-ranging NHPs, while parasites transmitted through either close and non-close contact, including the fecal–oral transmission, are more likely detected in captive NHPs [114].

Regarding the diagnostic method, morphological approaches were found to be the most used, followed by molecular procedures.

The most common biological samples were blood and stool, and ectoparasites corresponded to the least reported. Sampling NHPs, specially free-ranging, is logistically challenging as invasive sampling techniques such as the collection of blood, requires field anesthesia; therefore, optimization of non-invasive surveillance on NHPs is critical for understanding disease ecology of pathogens and identifying zoonotic diseases likely to emerge [115]. In this context, non-invasive methods such as stool collection are among the safest alternatives to study multiple aspects of the biology of NHPs [2]. However, even the collection of stool samples requires considerable efforts for their assignment to a specific individual, as well as to avoid multiple sampling for the same individual and later calculate the prevalence of parasites. 

Parasitological surveys of NHPs contribute to the understanding of the epidemiology, zoonotic emergence risk and transmission dynamics [41]. In this context, parasitological studies using adequate tools to evaluate the zoonotic potential are necessary. In the present review, as most studies are based on parasite morphology, some parasite species and/or genetic variants could not be determined, thus not allowing to assess their zoonotic potential. In future studies, the use of molecular tools will become essential, not only to identify and determine the presence/absence of parasites, but also to identify species/variants of the parasites circulating in each NHPs species and in each sampling site in order to better understand their distribution in NHPs and to evaluate transmission dynamics. Although there are challenges related to the molecular processing of the samples (e.g., disruption of the *Ascaris* and *Trichuris* eggshells prior to DNA extraction), efforts should be made to develop efficient protocols especially in stool samples. These are considered reliable for the non-invasive detection of pathogens, opening up new possibilities in the molecular epidemiology and evolutionary analysis of infectious diseases [2].

Molecular tools have been mainly used in studies aimed to detect protozoans: *Plasmodium* sp., *Toxoplasma* sp., *Entamoeba* sp., *Giardia* sp., *Blastocystis* sp., *Leishmania* sp., *Trypanosoma* sp., and *Pentatrichomonas* sp. [25,42,45,64,65,76,85]. However, the determination of parasite subtypes/genetic lineages was performed in few studies. Some studies on *Blastocystis* assessed the genetic variability and host specificity, reporting different subtypes (ST1-ST5, ST8) [42,61,116]. Studies on *Trypanosoma cruzi* identified the genetic lineages of the parasite (TcI-TcIII, TcV, TcVI) [64,90] as well as types of *Toxoplasma gondii* (Type I, Type II, non-archetypal) [39,100]. Molecular approaches were less frequent in studies of nematodes: *Trypanoxyuris* sp., *Dipetalonema* sp., *Mansonella* sp., *Brugia* sp., *Pterygodermatites* sp. [26,59,102,117], cestodes: *Mesocestoides* sp. [96], and ectoparasites: *Amblyomma* sp. [109]. No molecular records on zoonotic parasites as *Trichuris* sp., *Ascaris* sp., *Cryptosporidium* sp., *Hymenolepis* sp., *Taenia* sp., *Strongyloides* sp., *Capillaria* sp., nor *Balantidium* sp. were found. 

As habitat loss and forest fragmentation are currently a concerning global trend, and NHPs are in closer contact with humans, consequent ecological changes need to be monitored. Forest fragmentation is one of the main factors threatening NHPs [8], affecting all but not only the six Neotropical species included into the World’s 25 Most Endangered NHPs List [112]. However, only two studies included in the present review accounted for forest fragmentation as a variable during the analyses [25,45], even if some were performed in fragmented areas [67,81,85,91,105]. We strongly encourage the inclusion of this crucial factor as a variable for future studies as a way of better understanding parasite ecology, taking into account that some studies had reported a higher parasite prevalence in NHPs living in fragmented habitats [62,118], while other authors had found a lower presence of parasites [69], in comparison to prevalence found in continuous forests. Additionally, parasite taxa composition may vary according to NHPs living condition [118,119]. 

Not only more efforts aimed to broaden the knowledge of parasites infecting NHPs are required but we also suggest the standardization of the result presentation/display. For instance, it is necessary to include the coordinates of the sampling sites and show information separately for each NHPs species and study site when sampling is simultaneously carried out in different sites, involving more than one NHPs species. Therefore, the availability of the necessary information to perform meta-analysis, spatial analyses, and calculate parasite prevalence is facilitated, allowing to draw conclusions usable to a better understanding of infection patterns. 

## 5. Conclusions

In the present review, parasitological records for 20 genera of NHPs mainly conducted on captive animals were retrieved. Morphological approaches were found to be the most used, and Protozoa was the most frequent parasite group reported. Parasitological studies on American NHPs still need to be performed, especially for some genera and species with several information gaps, as well as Critically Endangered and Endangered primates, in both free-ranging and captive conditions. Parasitological studies using adequate tools to evaluate potential zoonoses are necessary in order to better understand the distribution of parasites in NHPs and to evaluate transmission dynamics, also taking considering factors as habitat loss and forest fragmentation.

## Figures and Tables

**Figure 1 microorganisms-09-02546-f001:**
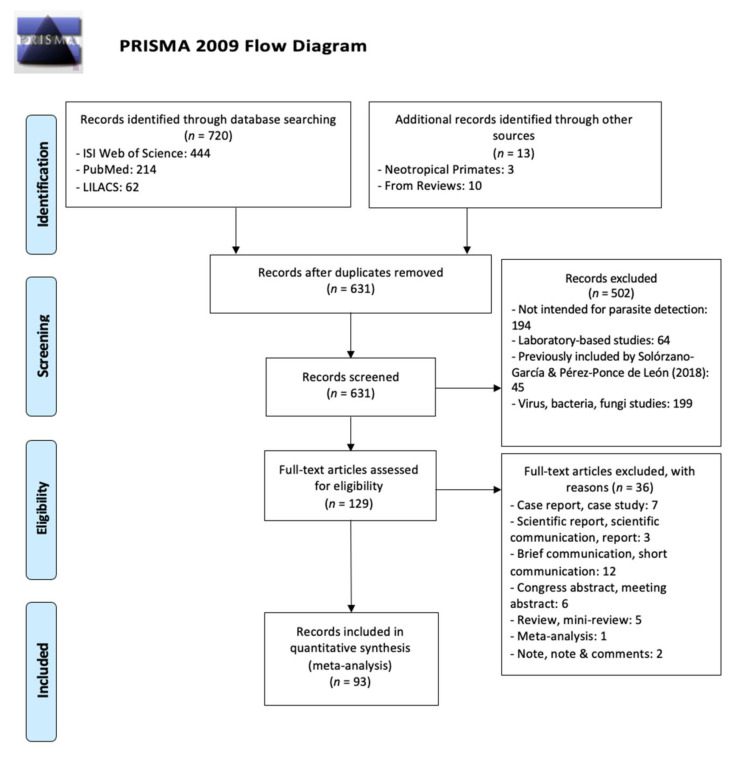
PRISMA Flow Diagram.

**Figure 2 microorganisms-09-02546-f002:**
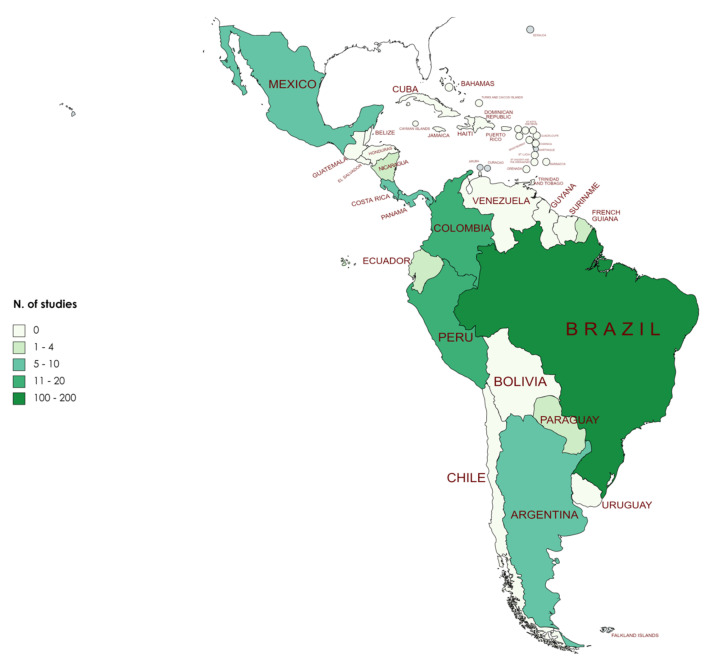
Geographical distribution of parasitological studies and number of studies per country and non-human primate genus.

**Table 1 microorganisms-09-02546-t001:** Number of articles per parasite group and non-human primate genus.

Non-Human Primate Genus	*n* Non-Human Primate Species Described in the Genus	*n* Non-Human Primate Species Studied	*n* Studies	Parasite Group					
				Protozoa	Trematoda	Cestoda	Nematoda	Acanthocephala	Ectoparasites
*Alouatta*	12	8	51	37	6	5	15	1	5
*Aotus*	11	7	17	15	1	0	2	0	1
*Ateles*	7	7	15	13	0	0	2	0	0
*Brachyteles*	2	2	4	4	0	0	0	0	0
*Cacajao*	3	2	3	2	0	0	1	0	0
*Callicebus*	5	3	9	7	0	0	0	0	2
*Callimico*	1	1	1	0	0	0	1	0	0
*Callithrix*	5	5	22	17	4	0	2	2	0
*Cebuella*	1	1	2	1	0	0	1	0	0
*Cebus*	14	6	11	10	0	2	2	1	0
*Cheracebus*	6	2	1	1	0	0	0	0	0
*Chiropotes*	5	3	7	7	0	0	0	0	0
*Lagothrix*	5	4	12	11	0	2	3	1	0
*Leontopithecus*	4	3	14	12	0	1	1	2	0
*Mico*	5	4	4	4	0	0	0	0	0
*Pithecia*	5	4	9	7	0	0	2	0	0
*Plecturocebus*	23	5	3	3	0	0	0	0	0
*Saguinus*	15	12	18	15	1	2	2	2	0
*Saimiri*	7	3	13	12	0	0	2	1	0
*Sapajus*	8	8	19	18	2	1	4	1	0

**Table 2 microorganisms-09-02546-t002:** Number of studies according to non-human primate living condition (captive/free-ranging), type of biological sample collected, and diagnostic method, per non-human primate genus.

Non-Human Primate Genus	% Non-Human Primate Living Condition (*n* Studies)		% Biological Sample (*n* Studies)					% Diagnostic Method (*n* Studies)		
	Free-Ranging	Captive	Blood	Serum	Stool	Tissue	Ectoparasites	Molecular	Morphological	Other *
*Alouatta*	69.6 (39)	30.4 (17)	27.1 (16)	11.9 (7)	39 (23)	13.6 (8)	8.5 (5)	41.9 (26)	43.5 (27)	14.5 (9)
*Aotus*	29.4 (5)	70.6 (12)	33.3 (7)	23.8 (5)	23.8 (5)	14.3 (3)	4.8 (1)	27.3 (6)	45.5 (10)	27.3 (6)
*Ateles*	38.9 (7)	61.1 (11)	29.4 (5)	17.6 (3)	47.1 (8)	5.9 (1)	0	50 (9)	33.3 (6)	16.7 (3)
*Brachyteles*	50 (2)	50 (2)	75 (3)	0	25 (1)	0	0	40 (2)	60 (3)	0
*Cacajao*	100 (3)	0	66.7 (2)	0	0	33.3 (1)	0	33.3 (1)	66.7 (2)	0
*Callicebus*	22.2 (2)	77.8 (7)	30 (3)	20 (2)	20 (2)	10 (1)	20 (2)	20 (2)	50 (5)	30 (3)
*Callimico*	0	100 (1)	0	0	100 (1)	0	0	50 (1)	50 (1)	0
*Callithrix*	54.2 (13)	45.8 (11)	35.7 (10)	21.4 (6)	17.9 (5)	25 (7)	0	28.9 (11)	47.4 (18)	23.7 (9)
*Cebuella*	50 (1)	50 (1)	50 (1)	0	50 (1)	0	0	0	100 (2)	0
*Cebus*	42.9 (6)	57.1 (8)	33.3 (4)	25 (3)	33.3 (4)	8.3 (1)	0	23.1 (3)	46.2 (6)	30.8 (4)
*Cheracebus*	100 (1)	0	100 (1)	0	0	0	0	0	100 (1)	0
*Chiropotes*	28.6 (2)	71.4 (5)	55.6 (5)	22.2 (2)	0	22.2 (2)	0	33.3 (4)	41.7 (5)	25 (3)
*Lagothrix*	35.7 (5)	64.3 (9)	25 (3)	25 (3)	41.7 (5)	8.3 (1)	0	23.1 (3)	53.8 (7)	23.1 (3)
*Leontopithecus*	50 (7)	50 (7)	33.3 (5)	33.3 (5)	20 (3)	13.3 (2)	0	22.2 (4)	38.9 (7)	38.9 (7)
*Mico*	25 (1)	75 (3)	50 (2)	0	50 (2)	0	0	40 (2)	60 (3)	0
*Pithecia*	33.3 (3)	66.7 (6)	44.4 (4)	22.2 (2)	11.1 (1)	22.2 (2)	0	30 (3)	50 (5)	20 (2)
*Plecturocebus*	50 (2)	50 (2)	100 (3)	0	0	0	0	50 (2)	50 (2)	0
*Saguinus*	29.4 (5)	70.6 (12)	45 (9)	25 (5)	20 (4)	10 (2)	0	35.7 (10)	42.9 (12)	21.4 (6)
*Saimiri*	25 (4)	75 (12)	38.5 (5)	23.1 (3)	23.1 (3)	15.4 (2)	0	47.8 (11)	34.8 (8)	17.4 (4)
*Sapajus*	44 (11)	56 (14)	41.7 (10)	20.8 (5)	20.8 (5)	16.7 (4)	0	35.5 (11)	41.9 (13)	22.8 (7)

* Other: Serology, ELISA, indirect ELISA, indirect agglutination assays, Western blood IgG assays, immunochromatographic assays, sero K-SeT rapid diagnostic tests, indirect immunofluorescence assays, immunohistochemical assays, antigen-based rapid diagnostic tests, TESA-blot.

**Table 3 microorganisms-09-02546-t003:** Parasites reported per non-human primate species. Parasites’ names were included exactly as reported in the retrieved publications.

Host	Parasite Group	Parasite Taxa	Parasites With Zero Prevalence *	References
Family Aotidae				
*Aotus*				
*Aotus* sp.	Protozoa	*Entamoeba coli, E. histolytica, Toxoplasma gondii,* Trypanosomatidae	*Trypanosoma cruzi*	[19,20,21,22]
*Aotus azarae*	Protozoa	*Trypanosoma cruzi*	*Plasmodium* sp.	[23,24]
*Aotus griseimembra*	Protozoa	*Plasmodium malariae/brasilianum*	*Plasmodium falciparum, P. vivax/simium*	[25]
	Nematoda		*Pterygodermatites nycticebi*	[26]
*Aotus infulatus*	Protozoa		*Leishmania sp., Plasmodium* sp., *P. berghei, P. brasilianum/malariae, P. falciparum, P. malariae, P. vivax*	[23,27,28,29,30]
*Aotus nancymaae*	Diptera	*Cuterebra* sp.		[31]
*Aotus nigriceps*	Protozoa	*Balantioides* sp., *Entamoeba* sp.	*Leishmania infantum*, *Plasmodium* sp., *Toxoplasma gondii, Trypanosoma cruzi*	[22,23,32,33,34]
	Trematoda	Trematoda		[34]
	Nematoda	Ascarididae, Strongylidae, *Strongyloides* sp., *Trypanoxyuris* sp.		[34]
*Aotus trivirgatus*	Protozoa	*Toxoplasma gondii, Leishmania braziliensis*	*Giardia* sp.	[32,35,36]
*Aotus vociferans*	Diptera	*Cuterebra* sp.		[31]
Family Atelidae				
*Alouatta*				
*Alouatta* sp.	Protozoa	*Blastocystis* sp., *Plasmodium vivax/simium, Toxoplasma gondii*		[20,21,37,38,39]
	Nematoda	*Trypanoxyuris minutus*		[40]
*Alouatta belzebul*	Protozoa	*Plasmodium brasilianum*	*Trypanosoma cruzi*	[23,24]
*Alouatta caraya*	Protozoa	*Blastocystis* sp., *B. hominis*, *Cryptosporidium* sp., *Eimeria* sp., *Entamoeba coli*, *Giardia* sp., *G. duodenalis*, *G. lamblia*, *Plasmodium brasilianum*, *P. falciparum*, *P. malariae*, *P. malariae/brasilianum*, *P. vivax*, *Leishmania amazonensis*, *L. braziliensis*, *L. infantum*	*Toxoplasma gondii*, *Trypanosoma cruzi*	[23,24,32,33,35,41,42,43,44,45,46]
	Cestoda	*Bertiella* sp., *B. mucronata*		[43,44,47]
	Nematoda	*Strongyloides* sp.		[43]
*Alouatta guariba*	Acanthocephala	*Pachysentis* sp.		[48]
	Protozoa	*Leishmania infantum*, *Blastocystis* sp., *Giardia* sp., *Plasmodium* sp., *P. malariae*, *P. brasilianum/malariae*, *P. simium*, *P. vivax*, *P. vivax/simium*,	*Toxoplasma gondii*	[21,23,29,30,32,35,46,49,50,51,52,53,54]
	Cestoda	*Bertiella* sp.		[48,55]
	Nematoda	*Ascaris* sp., *Trypanoxyuris minutus*		[48,55,56]
	Ixodida	*Amblyomma aureolatum*, *A. dubitatum*, *A. parkeri*, *A. sculptum*		[57]
	Phthiraptera	*Cebidicola semiarmatus*		[58]
	Siphonaptera	*Tunga penetrans*		[55]
*Alouatta macconnelli*	Protozoa		*Trypanosoma cruzi*	[24]
	Nematoda	*Brugia* sp., *Mansonella* sp.		[59]
*Alouatta palliata*	Protozoa	*Balantidium* sp., *Blastocystis* sp., *Entamoeba* sp., *Entamoeba/Endolimax* sp., *Cyclospora* sp., *Isospora* sp., *Iodamoeba* sp., *Dientamoeba* sp., *Chilomastix* sp., *Giardia* sp., *Toxoplasma gondii*, *Trichomonas* sp., *Trypanosoma cruzi*,		[60,61,62,63,64,65]
	Trematoda	*Controrchis* sp.		[60,62]
	Nematoda	*Capillaria* sp., *Enterobius* sp., *Trypanoxyuris* sp., *T. minutus*, *T. multilabiatus*, *Strongyloides* sp.		[60,62,66]
*Alouatta pigra*	Protozoa	*Blastocystis* sp., *Entamoeba* sp., *E. coli*, *Trypanosoma cruzi*	*Leishmania* sp.	[61,64,65,67,68]
	Trematoda	*Controrchis* sp., *C. biliophilus*		[67,69,70]
	Nematoda	*Parabronema* sp., *Trypanoxyuris* sp.		[67,69,70]
*Alouatta sara*	Protozoa	*Tetratrichomonas* sp.		[71]
*Alouatta seniculus*	Protozoa	*Balantidium coli*, *Blastocystis* sp., *Endolimax nana*, *Entamoeba coli*, *E. histolytica*, *Plasmodium falciparum*, *P. malariae/brasilianum*, *P. brasilianum*, *P. vivax/simium*, *Trypanosomatidae*, *Leishmania* sp., *L. guyanensis*, *L. infantum*	*Trypanosoma cruzi*	[19,22,23,25,42,72]
	Nematoda	*Ascaris lumbricoides*, *Enterobius vermicularis*, *Trypanoxyuris kemuimae*, *T. kotudoi*, *T. seunimii*, *Strongyloides* sp.		[19,73]
	Ixodida	*Rhipicephalus microplus*		[74]
*Ateles*				
*Ateles* sp.	Protozoa	*Giardia* sp., *Toxoplasma gondii*		[20,35]
*Ateles belzebuth*	Protozoa	*Blastocystis* sp., Trypanosomatidae	*Trypanosoma cruzi*	[22,42]
*Ateles chamek*	Protozoa	*Plasmodium brasilianum*, Trypanosomatidae	*Giardia* sp., *Tetratrichomonas* sp., *Trypanosoma cruzi*	[22,23,35,71]
*Ateles fusciceps*	Protozoa	*Blastocystis* sp.	*Leishmania* sp.	[42,68]
*Ateles geoffroyi*	Protozoa	*Toxoplasma gondii*, *Trypanosoma cruzi*		[63,64]
	Nematoda	*Trypanoxyuris atelis*, *T. atelophora*		[40,66]
*Ateles hybridus*	Protozoa	*Plasmodium malariae/brasilianum*, *P. vivax/simium*	*Plasmodium falciparum*	[25]
*Ateles marginatus*	Protozoa		*Giardia* sp., *Plasmodium* sp., *Tetratrichomonas* sp.	[23,35,71]
*Ateles paniscus*	Protozoa	*Cryptosporidium parvum*, *Trypanosoma cruzi*, *Blastocystis* sp., *Plasmodium brasilianum*	*Giardia* sp.	[23,24,35,42,75,76]
*Brachyteles*				
*Brachyteles arachnoides*	Protozoa	*Giardia* sp., *Plasmodium brasilianum*, *P. simium*		[23,35]
*Brachyteles hypoxanthus*	Protozoa		*Plasmodium* sp., *Leishmania* sp.	[29,30,52]
*Lagothrix*				
*Lagothrix* sp.	Protozoa		*Toxoplasma gondii*	[20]
*Lagothrix cana*	Protozoa	*Plasmodium brasilianum*, *Trypanosomatidae*, *Trypanosoma cruzi*	*Toxoplasma gondii*, *Leishmania infantum*	[22,23,24,32,33]
*Lagothrix flavicauda*	Acanthocephala	*Prosthenorchis elegans*		[77]
	Protozoa	*Cryptosporidium* sp., *Entamoeba coli*		[77]
	Cestoda	*Paratriotaenia oedipomidatis*		[77]
	Nematoda	*Ancylostoma* sp., *Capillaria* sp., *Strongyloides* sp., *S. cebus*		[77]
*Lagothrix lagotricha*	Protozoa	*Blastocystis* sp., Entamoebidae, *Giardia* sp., Trypanosomatidae	*Tetratrichomonas* sp., * Trypanosoma cruzi*	[22,23,35,42,71,78]
	Cestoda	Cestoda		[78]
	Nematoda	Ascarididae, Oxyuridae, Strongylidae, Trichinellidae, Trichostrongylidae		[78]
*Lagothrix poeppigii*	Protozoa	Trypanosomatidae, *Trypanosoma cruzi*		[22]
	Nematoda	*Dipetalonema gracile*		[79]
Family Callithrichidae				
*Callimico*				
*Callimico goeldii*	Nematoda	*Pterygodermatites nycticebi*		[26]
*Callithrix*				
*Callithrix* sp.	Acanthocephala	*Pachysentis* sp.		[48]
	Protozoa	*Cryptosporidium parvum*, *Toxoplasma gondii*, *Leishmania braziliensis/infantum/amazonensis*, *L. infantum*	*Giardia* sp., *Plasmodium brasilianum*, *P. falciparum*, *P. malariae/brasilianum*, *P. vivax/simium*, *Trypanosoma cruzi*	[20,21,35,52,54,75,80,81]
	Trematoda	*Platysosomum* sp.		[48]
	Nematoda	*Trypanoxyuris callithricis*, *Primasubulura* sp.		[48]
*Callithrix aurita*	Protozoa		*Plasmodium* sp.	[23]
*Callithrix geoffroyi*	Acanthocephala	*Prosthenorchis elegans*		[82]
	Protozoa	*Plasmodium brasilianum*, *Trypanosoma minasense*	*Blastocystis* sp., *Plasmodium falciparum*, *P. vivax/simium*	[23,42,80,83]
	Trematoda	*Platysosomum* sp.		[84]
	Nematoda	*Gongylonema* sp.		[84]
*Callithrix jacchus*	Protozoa	*Blastocystis* sp., *Plasmodium* sp., *Leishmania* sp.	*Giardia* sp., * Plasmodium berghei*, *P. brasilianum*, *P. brasilianum/malariae*, *P. falciparum*, *P. vivax*, *P. vivax/simium*, *Tetratrichomonas* sp.	[23,27,28,35,42,71,80,85]
*Callithrix kuhlii*	Protozoa		*Giardia* sp.	[35]
*Callithrix penicillata*	Protozoa	*Giardia* sp., *Tetratrichomonas* sp., *Leishmania* sp., *Trypanosoma minasense*	*Plasmodium* sp.	[23,35,46,71,85,86]
	Trematoda	*Platynosomum illicens*		[87,88]
*Cebuella*				
*Cebuella pygmaea*	Protozoa		*Plasmodium* sp.	[23]
	Nematoda		*Pterygodermatites nycticebi*	[26]
*Leontopithecus*				
*Leontopithecus* sp.	Protozoa	*Toxoplasma gondii*		[20]
*Leontopithecus chrysomelas*	Acanthocephala	*Prosthenorchis* sp.		[48,89]
	Protozoa	*Blastocystis* sp., *Plasmodium brasilianum*, *Trypanosoma cruzi*	*Giardia* sp., *Plasmodium* sp., *P. falciparum*, *P. vivax/simium*, *Toxoplasma gondii*, *Leishmania infantum*	[32,33,35,42,80,90,91]
	Nematoda	*Trypanoxyuris* sp., Spiruridae, *Primasubulura* sp.		[89]
*Leontopithecus chrysopygus*	Protozoa	*Giardia* sp.	*Plasmodium* sp., *P. brasilianum*, *P. falciparum*, *P. vivax/simium*, *Toxoplasma gondii*, *Leishmania infantum*	[32,33,35,80]
*Leontopithecus rosalia*	Protozoa	*Giardia* sp., *Plasmodium brasilianum*, *Leishmania infantum*, *Toxoplasma gondii*, *Trypanosoma cruzi*	*Blastocystis* sp., *Plasmodium* sp., *P. falciparum, P. vivax/simium,*	[23,29,30,32,33,35,42,52,80,90,92]
	Cestoda	Cestoda		[48]
*Mico*				
*Mico argentatus*	Protozoa	*Blastocystis* sp.	*Giardia* sp.	[35,42]
*Mico emiliae*	Protozoa	*Plasmodium* sp.		[23]
*Mico humeralifer*	Protozoa	*Plasmodium brasilianum*	*Plasmodium* sp., *P. falciparum*, *P. vivax/simium*	[23,80]
*Mico mauesi*	Protozoa		*Plasmodium* sp., *P. brasilianum*, *P. falciparum*, *P. vivax/simium*	[80]
*Saguinus*				
*Saguinus* sp.	Protozoa	*Plasmodium brasilianum*	*Plasmodium falciparum*, *P. vivax/simium*, *Toxoplasma gondii*	[20,80]
*Saguinus bicolor*	Protozoa		Blastocystis sp., *Plasmodium brasilianum*, *P. falciparum*, *P. vivax/simium*, *Trypanosoma cruzi*	[23,24,42,80]
*Saguinus fuscicollis*	Protozoa	*Plasmodium brasilianum*	Blastocystis sp., *Giardia* sp., *Plasmodium* sp., *Trypanosoma cruzi*	[22,23,35,42,93]
*Saguinus geoffroyi*	Protozoa		*Leishmania braziliensis*	[68]
*Saguinus imperator*	Acanthocephala	*Prosthenorchis* sp.		[94]
	Protozoa	*Plasmodium brasilianum*	*Toxoplasma gondii*, *Leishmania* sp., *L. infantum*	[23,29,30,32,33,93]
	Trematoda	Dicrocoeliidae		[94]
	Cestoda	Cestoda		[94]
	Nematoda	Gongylonematidae, *Primasubulura* sp.		[94]
*Saguinus labiatus*	Protozoa		*Plasmodium* sp.	[23]
*Saguinus leucopus*	Acanthocephala	*Prosthenorchis* sp.		[95]
	Protozoa	*Endolimax* sp., *Entamoeba* sp., *Trypanosoma* sp.		[95]
	Nematoda	Nematoda, Ancylostomatidae, *Ascaris* sp., Metastrongylidae, Spiruridae, *Strongyloides* sp., *Trichostrongylus* sp.		[95]
*Saguinus martinsi*	Protozoa	*Plasmodium brasilianum*	*Plasmodium falciparum, P. vivax/simium*	[80]
*Saguinus midas*	Protozoa	*Trypanosoma cruzi, Plasmodium brasilianum*	*Giardia* sp., *Plasmodium* sp., *P. berghei*, *P. brasilianum/malariae*, *P. falciparum*, *P. malariae*, *P. vivax*, *P. vivax/simium*	[23,24,27,28,35,80]
	Cestoda	*Mesocestoides* sp.		[96]
*Saguinus mystax*	Protozoa		*Plasmodium* sp.	[23]
*Saguinus niger*	Protozoa		*Plasmodium brasilianum*, *P. falciparum*, *P. vivax/simium*	[80]
*Saguinus oedipus*	Protozoa		*Giardia* sp.	[35]
*Saguinus weddelli*	Acanthocephala	*Prosthenorchis* sp.		[94]
	Protozoa	*Blastocystis* sp.		[42]
	Trematoda	Dicrocoeliidae		[94]
	Cestoda	Cestoda		[94]
	Nematoda	*Primasubulura* sp., Gongylonematidae		[94]
Family Cebidae				
*Cebus*				
*Cebus* sp.	Protozoa	*Toxoplasma gondii*		[20]
*Cebus albifrons*	Acanthocephala	*Prosthenorchis elegans*		[97]
	Protozoa	*Entamoeba coli*, *E. histolytica*, *E. histolytica/dispar/moskovskii/nuttallli*, *Trypanosomatidae*, *Trypanosoma cruzi*	*Plasmodium* sp.	[19,22,23,24,97]
	Cestoda	*Hymenolepis* sp.		[97]
	Nematoda	*Capillaria* sp., *Enterobius vermicularis*, Strongyles, *Strongyloides* sp.		[19,97]
*Cebus capucinus*	Protozoa		*Leishmania* sp.	[68]
	Cestoda		*Echinococcus* sp., *Taenia* sp.	[98]
*Cebus imitator*	Protozoa	*Toxoplasma gondii*		[63]
*Cebus kaapori*	Protozoa	*Giardia* sp.		[35]
*Cebus olivaceus*	Protozoa	*Trypanosoma cruzi*	*Giardia* sp., *Plasmodium* sp.	[23,24,35]
*Cebus versicolor*	Protozoa	*Plasmodium malariae/brasilianum, P. vivax/simium*	*Plasmodium falciparum*	[25]
*Saimiri*				
*Saimiri* sp.	Protozoa	*Toxoplasma gondii*	*Tetratrichomonas* sp.	[20,71]
*Saimiri boliviensis*	Protozoa	*Plasmodium brasilianum*, Trypanosomatidae, *Trypanosoma cruzi*		[22,23]
*Saimiri sciureus*	Acanthocephala	*Prosthenorchis elegans*		[99]
	Protozoa	*Blastocystis* sp., *Giardia* sp., *Plasmodium brasilianum*, *Toxoplasma gondii,* Trypanosomatidae, *Trypanosoma cruzi*	*Plasmodium* sp., *P. berghei, P. brasilianum/malariae, P. falciparum*	[22,23,24,27,28,35,76,99,100,101]
	Nematoda	*Ancylostoma* sp., *Dipetalonema gracile*, Oxyuridae, *Strongyloides* sp.		[99,102]
*Saimiri ustus*	Protozoa	*Plasmodium brasilianum*, *Trypanosoma cruzi*		[23,24]
*Sapajus*				
*Sapajus* sp.	Protozoa	*Plasmodium* sp., *P. brasilianum/malariae*, *P. falciparum*, *P. malariae*, *Toxoplasma gondii*, *Trypanosoma cruzi*	*Plasmodium berghei*, *P. vivax*	[21,24,27,28]
*Sapajus apella*	Protozoa	*Blastocystis* sp., *Plasmodium falciparum*, *P. malariae/brasilianum*, *P. brasilianum*, *P. vivax*, *Toxoplasma gondii*, *Tetratrichomonas* sp., *Leishmania infantum*, *L. shawi*, *Trypanosoma cruzi*	*Giardia* sp.	[21,23,24,32,33,35,42,46,71,76]
	Nematoda	*Molineus torulosus*		[103]
*Sapajus cay*	Protozoa	*Leishmania* sp.	*Trypanosoma cruzi*, *T. evansi*	[104]
*Sapajus flavius*	Protozoa	*Leishmania* sp., *Trypanosoma cruzi*, *Plasmodium* sp., *Toxoplasma gondii*	*Giardia* sp.	[35,105]
	Nematoda	Nematoda, *Molineus torulosus*		[103,105]
*Sapajus libidinosus*	Protozoa		*Plasmodium* sp.	[23]
	Nematoda	*Molineus torulosus*		[103]
*Sapajus macrocephalus*	Protozoa	*Plasmodium brasilianum,* Trypanosomatidae, *Trypanosoma cruzi*		[22,23,24]
*Sapajus nigritus*	Acanthocephala	*Prostenorchis* sp.		[48]
	Protozoa	*Toxoplasma gondii*	*Plasmodium* sp.	[21,23,52]
	Trematoda	Trematoda, *Platynosomum* sp.		[48,106]
	Cestoda	Hymenolepididae		[106]
	Nematoda	*Ascaris* sp., *Filariopsis* sp., *Physaloptera* sp., *Strongyloides* sp., Subuluridae, *Trichuris* sp.		[48,106]
*Sapajus robustus*	Protozoa		*Plasmodium* sp.	[23]
*Sapajus xanthosternos*	Protozoa		*Giardia* sp., *Leishmania* sp.	[29,30,35]
Family Pitheciidae				
*Cacajao*				
*Cacajao calvus*	Protozoa	*Plasmodium brasilianum*, Trypanosomatidae	*Trypanosoma cruzi*	[22,23]
	Nematoda	*Dipetalonema freitasi*		[107]
*Cacajao melanocephalus*	Protozoa		*Plasmodium* sp.	[23]
*Callicebus*				
*Callicebus melanochir*	Protozoa		*Plasmodium* sp.	[23]
*Callicebus nigrifrons*	Protozoa	*Cryptosporidium* sp., *Toxoplasma gondii*	*Blastocystis* sp., *Plasmodium* sp., *Leishmania infantum*	[23,32,33,42,52,108]
	Ixodida	*Amblyomma parkeri*, *A. romarioi*		[57,109]
*Callicebus personatus*	Protozoa		*Leishmania* sp., *Plasmodium* sp.	[23,29,30]
*Chiropotes*				
*Chiropotes albinasus*	Protozoa	*Plasmodium brasilianum*		[23]
*Chiropotes chiropotes*	Protozoa	*Plasmodium brasilianum*	*Trypanosoma cruzi*	[23,24]
*Chiropotes satanas*	Protozoa	*Plasmodium brasilianum*, *Toxoplasma gondii*, *Leishmania shawi*	*Plasmodium* sp., *P. berghei*, *P. brasilianum/malariae*, *P. falciparum*, *P. vivax*, *Leishmania* sp.	[23,27,28,29,30,32,110]
*Pithecia*				
*Pithecia albicans*	Protozoa		*Giardia* sp.	[35]
*Pithecia irrorata*	Protozoa		*Plasmodium* sp., *Toxoplasma gondii, Leishmania* sp., *L. infantum*	[23,29,30,32,33]
*Pithecia monachus*	Protozoa	*Plasmodium brasilianum*, Trypanosomatidae, *Trypanosoma cruzi*		[22,23,24]
	Nematoda	*Dipetalonema gracile*, *Strongyloides stercoralis*		[107,111]
*Pithecia pithecia*	Protozoa	*Plasmodium brasilianum*	*Giardia* sp.	[23,35]

* Studies were conducted in order to detect those parasites, but zero prevalence was reported.

## Data Availability

Not applicable.

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
