# Peer review of "Parasites of Free-Ranging and Captive American Primates: A Systematic Review"

_microorganisms, 2021, doi:10.3390/microorganisms9122546_

Round 1

Reviewer 1 Report

In this systematic review, the authors compile an overview of the parasites of free-ranging and captive American primates to identify information gaps, support decision making and lay bare leads for future research projects.

The systematic review follows the academic and international standard (PRISMA) and the methodology is well documented, the choices made in terms of databases and search criteria are clear and reasonable. The authors give detailed methodological information such as languages of the literature surveys as part of the systematic review, type of studies and type of parasites included. The standardized form for data extraction is comprehensive and well designed.

In Figure 1, the PRISMA Flow Diagram would be more informative if:

  • Reasons were given for excluding 502 records from the 631 screened as it is a very high number
  • Under the full text articles excluded with reason behind each reason the number of actual exclusions would be mentioned

On page 5, the discussion surrounding the records is difficult to follow. The authors argue that “there were records for captive Neotropical primates in Asia and Europe:, but it is not clear whether records is another word for publications (n= 93) post June 2017, which were extensively reviewed or whether ‘records’ refers to something different. It appears entirely plausible that there may be no recent (post 2017) records from a number of South and Central American Countries and limited records from Asia and Europe, but the information the authors provide in the supplementary material refers back to records as old as 1879, 1903 etc. Some of the references provided in the supplementary table [12, 13, 16, 17 etc.] are from before June 2017, which the authors have taken as the start of their window of enquiry. Unless this is clearly explained, it generates major confusion.

Also, certain records presented in the supplementary material without any comments, which also generates confusion. The authors of the review present some parasites of the genus Plasmodiidae first as separate: P. brasilianum, P. malariae (or P. simium, P. vivax) and then as one P. brasilianum/P. malariae (or P. simium/P. vivax). This may be unclear to a reader if the authors do not explain that certain authors refer to P. brasilianum and P. malariae (or P. simium and P. vivax) as two separate parasite species; however, in view of the most recent molecular findings (Tazi, 2011), other authors consider these duos one species.

In table 1 it would be good to add a few lines about the detection of multiple parasites in one study. Are these multiple detections reliable e.g. was the detection method specific enough? A short comment on the reliability of the data included is necessary if the review is to support decision making, further research and to provide information of research gaps as the authors argue.

Table 2 and Table 3 are very clear and informative.

Figure 2 is clear and informative.

In the discussion, it is unclear whether publications were found or not concerning Alouatta guariba (line 150) as it is initially mentioned but not followed up on.

Cebus (C.) equatorialis is mentioned twice: once as part of the 25 most endangered NHP species (line 149) and once as part of the critically endangered NHP species (line 154).

Between lines 148 and 156, in the beginning of the discussion the authors argue that no records are available from several most endangered, critically endangered, and endangered species. In line 166 show that over half of the studies they reviewed were conducted on captive species, which may in part explain the finding that most endangered, critically endangered, and endangered species are understudies because it is less likely that they are kept captive. However, the authors do not point to this possible nexus.

Although other kinds of studies (e.g behavioral, genetic) may have been carried out for those species during the time range considered in this study, it must be highlighted that parasitological studies are also very important, and a useful insight for monitoring the health status, as disease is a threat for NHP.” (lines 156-159)

While this is an interesting argument, it looks rather superficial. The presence or absence of a specific parasite is not the same a monitoring the health status of the NHP as the health status is determined by the interaction between the specific parasite and the NHP. A number of wild NHP species can live with Plasmodium parasites perfectly and control the infection without problems, thus having a specific parasite does not mean automatically having disease manifestations and does not automatically form a threat for NHPs. In any given case it has to be understood whether e.g. due to human intervention the cases in a specific parasite have increased, whether this negatively affects the population of a specific NHP genus etc. Thus, in my view, the authors should advocate for example for correlation studies to understand the multifaceted relationships between the NHPs, their environment, their parasites and disease.

Between lines 174 and 182, the authors analyzed the number of records found per NHP genus. Again, there may be a correlation between NHP genus that are more frequently held in captivity and the finding of the authors, but the nexus is not mentioned.

In the present revision, as most studies are based on parasite morphology, parasite species or genetic variants could not be determined, therefore, zoonotic potential could not be properly determined.” (Lines 185-187)

Revision is not the best word, it should be called review

While in essence it is true that molecular tools are more informative than morphological studies, it is not correct in general that the zoonotic potential can only be properly determined by molecular tools as different parasite species e.g. different Plasmodium species have been identified far before the discovery of molecular tools.

Although the authors present findings related to the collection of samples, they do not discuss the issues related to the availability of samples. In free ranging NHPs in general only stool samples are available as to get blood or serum the animals would need to be captured and their natural habits would have to be disrupted with serious consequences for their welfare. It is not always possible to connect stool samples unless collected in tandem with very careful observations to the NHP species that excreted them and it is even more problematic to connect findings obtained in stool samples to the individual level, which may be necessary when looking at prevalence on the population level. It is also much more difficult to adapt molecular technologies to stool samples than to blood.

The term “primate” should in most cases be substituted with the dicit non-human primate (NHP) as it primate and NHP is not the same thing and should not be used interchangeably.

Minor issues:

Line 73: or have been focused in fragmentation on fauna living in the American 73 tropics [13,14].

Unclear. The sentence should likely read: or have been focused ON THE fragmentation of THE fauna living in the American tropics [13,14].

Line 86: using the terms ''parasite'' a primate genus (e.g. parasite AND Cebus).

Should be: using the terms ''parasite'' AND primate genus (e.g. parasite AND Cebus).

Figure 1 PRISMA Flow Diagram.

“Not intended to parasite detection” should be Not intended FOR parasite detection

Line 121: “while genera with less records were Callimico (n=1) and Cebuella (n=1)

This statement is not correct, the authors probably mean: while genera with the least records were Callimico (n=1) and Cebuella (m=1)

Line 148: “The most recent list (2018-2020) of the world’s 25 most ndangered NHP species

Should be: The most recent list (2018-2020) of the world’s 25 most ENDANGERED NHP species…

Line 175: “while for all other genera there were retrieved less than 20 records (Table 1)

Should be: while for all other genera LESS than 20 records WERE RETRIEVED (Table 1)

Author Response

Dear reviewer,

We are very grateful to you for your effort and dedication in revising our systematic review.

We really appreciate the insightful comments and suggestions, and we have addressed them point by point. All changes have been marked up with the “track changes” function, within the manuscript.

We hope that this version of the review has been considerably improved.

_________________________________________________________________________________

In Figure 1, the PRISMA Flow Diagram would be more informative if:

-Reasons were given for excluding 502 records from the 631 screened as it is a very high number

Authors: The PRISMA Flow Diagram was modified, including the requested information.

- Under the full text articles excluded with reason behind each reason the number of actual exclusions would be mentioned

Authors: The PRISMA Flow Diagram was modified, including the requested information.

On page 5, the discussion surrounding the records is difficult to follow. The authors argue that “there were records for captive Neotropical primates in Asia and Europe:”, but it is not clear whether records is another word for publications (n= 93) post June 2017, which were extensively reviewed or whether ‘records’ refers to something different. It appears entirely plausible that there may be no recent (post 2017) records from a number of South and Central American Countries and limited records from Asia and Europe, but the information the authors provide in the supplementary material refers back to records as old as 1879, 1903 etc. Some of the references provided in the supplementary table [12, 13, 16, 17 etc.] are from before June 2017, which the authors have taken as the start of their window of enquiry. Unless this is clearly explained, it generates major confusion.

Authors: Publications that are from before June 2017 correspond to studies taken from reviews, and that were not previously included in the review by Solórzano-García & Pérez-Ponce de León (2018). In the figure 1 those publications are indicated as “Additional records identified through other sources”. Additionally, the word “records” was changed to “publications”.

Also, certain records presented in the supplementary material without any comments, which also generates confusion. The authors of the review present some parasites of the genus Plasmodiidae first as separate: P. brasilianum, P. malariae (or P. simium, P. vivax) and then as one P. brasilianum/P. malariae (or P. simium/P. vivax). This may be unclear to a reader if the authors do not explain that certain authors refer to P. brasilianum and P. malariae (or P. simium and P. vivax) as two separate parasite species; however, in view of the most recent molecular findings (Tazi, 2011), other authors consider these duos one species.

Authors: We agree that this information may be unclear, however, parasites’ names were included exactly as reported in the publications. Aiming to clarify, we included this statement in the caption of Table 3 and in the Supplementary Material: “Parasites’ names were included exactly as reported in the retrieved publications.”

In table 1 it would be good to add a few lines about the detection of multiple parasites in one study. Are these multiple detections reliable e.g. was the detection method specific enough? A short comment on the reliability of the data included is necessary if the review is to support decision making, further research and to provide information of research gaps as the authors argue.

Authors: Table 1 does not report the number of parasite species detected but the number of studies for parasite groups at the higher taxonomic level. Although we agree that the reliability of findings may depend on the methodological approach, in this case we are confident that errors in their assignment at phylum or class level can be excluded.

Table 2 and Table 3 are very clear and informative.

Authors: We thank the reviewer for the positive comment

Figure 2 is clear and informative.

Authors: We thank the reviewer for the positive comment

In the discussion, it is unclear whether publications were found or not concerning Alouatta guariba (line 150) as it is initially mentioned but not followed up on.

Authors: We included the number of publications that we found regarding A. guariba in the Discussion.

Cebus (C.) equatorialis is mentioned twice: once as part of the 25 most endangered NHP species (line 149) and once as part of the critically endangered NHP species (line 154).

Authors: We removed “C. aequatorialis” when mentioned for the second time.

Between lines 148 and 156, in the beginning of the discussion the authors argue that no records are available from several most endangered, critically endangered, and endangered species. In line 166 show that over half of the studies they reviewed were conducted on captive species, which may in part explain the finding that most endangered, critically endangered, and endangered species are understudies because it is less likely that they are kept captive. However, the authors do not point to this possible nexus.

Authors: The following sentence has been added: “The amount of information is probably biased by the availability of different species in captivity, a condition that strongly facilitate parasitological investigations. It can be spec-ulated that the lack of information for endangered species could be related to their scarcity in captive conditions.

“Although other kinds of studies (e.g behavioral, genetic) may have been carried out for those species during the time range considered in this study, it must be highlighted that parasitological studies are also very important, and a useful insight for monitoring the health status, as disease is a threat for NHP.” (lines 156-159). While this is an interesting argument, it looks rather superficial. The presence or absence of a specific parasite is not the same a monitoring the health status of the NHP as the health status is determined by the interaction between the specific parasite and the NHP. A number of wild NHP species can live with Plasmodium parasites perfectly and control the infection without problems, thus having a specific parasite does not mean automatically having disease manifestations and does not automatically form a threat for NHPs. In any given case it has to be understood whether e.g. due to human intervention the cases in a specific parasite have increased, whether this negatively affects the population of a specific NHP genus etc. Thus, in my view, the authors should advocate for example for correlation studies to understand the multifaceted relationships between the NHPs, their environment, their parasites and disease.

Authors: Thank you for the observation, the phrase was modified as: “representing a useful insight for monitoring the health status of NHPs in contexts of human-NHPs interfaces, as human-induced forest loss increase the exposition of NHPs to human and domesticated animal pathogens [8]. Additionally, even if non-lethal parasite infections are common in wild NHPs, parasite infections could cause sickness behaviors that may be adaptative in the short term, but have longer-term fitness consequences [116].”

Between lines 174 and 182, the authors analyzed the number of records found per NHP genus. Again, there may be a correlation between NHP genus that are more frequently held in captivity and the finding of the authors, but the nexus is not mentioned.

Authors: Please refer to the answer provided above for the same comment. Moreover, as requested by Reviewer 3, the entire paragraph was placed into the results’ section: “Alouatta was the NHPs genus with most records (n=51) followed by Callithrix (n=22), while for all other genera less than 20 records were retrieved (Table 1). Considering the number of species, the genera with more species are Plecturocebus (n=23), Saguinus (n=15), Cebus (n=14), Alouatta (n=12), and Aotus (n=11). Likewise, Ateles, Brachyteles, Callimico, Callithrix, Cebuella, and Sapajus had records for 100% of the species, and 80% of the species for Lagothrix, Pithecia, and Saguinus. Brazil was the country with most parasitological records (n=163), being also the country with highest recorded occurrence of NHPs. Other NHPs rich-countries, such as Peru and Colombia, were the second and third countries with more records (20 and 13, respectively).”

“In the present revision, as most studies are based on parasite morphology, parasite species or genetic variants could not be determined, therefore, zoonotic potential could not be properly determined.” (Lines 185-187)

Revision is not the best word, it should be called review

Authors: We agree.

While in essence it is true that molecular tools are more informative than morphological studies, it is not correct in general that the zoonotic potential can only be properly determined by molecular tools as different parasite species e.g. different Plasmodium species have been identified far before the discovery of molecular tools.

Authors: Thank you for the comment. We agree and thus we modified the sentence as follows: “as most studies are based on parasite morphology, SOME parasite species AND/or genetic variants could not be determined, thus not allowing to assess their zoonotic potential”

Although the authors present findings related to the collection of samples, they do not discuss the issues related to the availability of samples. In free ranging NHPs in general only stool samples are available as to get blood or serum the animals would need to be captured and their natural habits would have to be disrupted with serious consequences for their welfare. It is not always possible to connect stool samples unless collected in tandem with very careful observations to the NHP species that excreted them and it is even more problematic to connect findings obtained in stool samples to the individual level, which may be necessary when looking at prevalence on the population level. It is also much more difficult to adapt molecular technologies to stool samples than to blood.

Authors: We modified the text as follows: “The most common biological samples were blood and stool, and ectoparasites corresponded to the least reported. Sampling NHPs, specially free-ranging, is logistically challenging as invasive sampling techniques such as the collection of blood, requires field anesthesia, therefore, optimization of non-invasive surveillance on NHPs is critical for understanding disease ecology of pathogens and identifying zoonotic diseases likely to emerge [116]. In this context, non-invasive methods such as stool collection are among the safest alternatives to study multiple aspects of the biology of NHPs [2]. However, even the collection of stool samples requires considerable efforts for their assignment to a specific individual, as well as to avoid multiple sampling for the same individual and later calculate the prevalence of parasites.”

Additionally, another part of the text was slightly modified as follows: “Although there are challenges related to the molecular processing of the samples (e.g dis-ruption of the Ascaris and Trichuris eggshells prior to DNA extraction), efforts should be made to develop efficient protocols especially in stool samples. These are considered reli-able for the non-invasive detection of pathogens, opening up new possibilities in the mo-lecular epidemiology and evolutionary analysis of infectious diseases [2].”

The term “primate” should in most cases be substituted with the dicit non-human primate (NHP) as it primate and NHP is not the same thing and should not be used interchangeably.

Authors: The term “primate” has been substituted along the manuscript.

Minor issues:

Line 73: “or have been focused in fragmentation on fauna living in the American 73 tropics [13,14].”

Unclear. The sentence should likely read: or have been focused ON THE fragmentation of THE fauna living in the American tropics [13,14].

Authors: Done.

Line 86: “using the terms ''parasite'' a primate genus (e.g. parasite AND Cebus).”

Should be: using the terms ''parasite'' AND primate genus (e.g. parasite AND Cebus).

Authors: Done.

Figure 1 PRISMA Flow Diagram.

“Not intended to parasite detection” should be Not intended FOR parasite detection

Authors: Done.

Line 121: “while genera with less records were Callimico (n=1) and Cebuella (n=1)”

This statement is not correct, the authors probably mean: while genera with the least records were Callimico (n=1) and Cebuella (m=1)

Authors: The sentence was changed as suggested.

Line 148: “The most recent list (2018-2020) of the world’s 25 most ndangered NHP species…” Should be: The most recent list (2018-2020) of the world’s 25 most ENDANGERED NHP species…

Authors: Done.

Line 175: “while for all other genera there were retrieved less than 20 records (Table 1)”. Should be: while for all other genera LESS than 20 records WERE RETRIEVED(Table 1)

Authors: Done.

Reviewer 2 Report

The review by Rondón et al. is well written and is an important contribution to covering the diversity of parasites in American primates. I consider it well suited for publishing in the journal Microorganisms.

I only have some minor comments referring to grammatical errors.

Minor comments:

Regarding the term NHP, I would write “NHPs” throughout the manuscript if multiple species are referred to.

L16: Change to “NHP (Platyrrhines) in the Americas”.

L121: Change “more records” to “most records”

L36: Change “world, as” to “world such as”

L72: Delete comma “… Plathyrrhines [5,12], or have been focused …”

L148: Change “ndangered” to “endangered”

L174: Change “more records” to “most records”

L176: Change “more records” to “most records”

L179: Change “more records” to “most records”

L180: Change “higher” to “the highest”

Author Response

 Dear reviewer,

We are very grateful to you for your effort and dedication in revising our systematic review.

We really appreciate the insightful comments, and we have addressed them point by point.

_______________________________________________________________________________

Minor comments:

Regarding the term NHP, I would write “NHPs” throughout the manuscript if multiple species are referred to.

Authors: Done.

L16: Change to “NHP (Platyrrhines) in the Americas”.

Authors: Done.

L121: Change “more records” to “most records”

Authors: Done.

L36: Change “world, as” to “world such as”

Authors: Done.

L72: Delete comma “… Plathyrrhines [5,12], or have been focused …”

Authors: Done.

L148: Change “ndangered” to “endangered”

Authors: Done.

L174: Change “more records” to “most records”

Authors: Done.

L176: Change “more records” to “most records”

Authors: Done.

L179: Change “more records” to “most records”

Authors: Done.

L180: Change “higher” to “the highest”

Authors: Done.

Reviewer 3 Report

The authors,

All comments and suggestions are given in the pdf file attached to this review. 
Although it has been a considerably tough work of reviewing the papers from the last 4 years and it is interesting the information provided about the parasites that have been found in Platyrrhines, there is a lot of work on the presentation of the results, especially for the table 3. 

I think the suggestion will considerably improve the manuscript for publication.

Regards,

Reviewer.

Author Response

Dear reviewer,

We are very grateful to you for your effort and dedication in revising our systematic review.

We really appreciate the insightful comments and suggestions, and we have addressed them point by point. All changes have been marked up with the “track changes” function, within the manuscript.

We hope that this version of the review has been considerably improved.
_________________________________________________________________________________

General comments.

  1. Lines 42-43: no matter how many manuscripts and books include viruses, bacteria and fungi into the “parasite” classification, they are not considered properly parasites. Therefore, I would suggest removing the word “parasitic” in line 42. Furthermore, in line 78 they talk about parasites, including protozoans, helminths and ectoparasites (which is correct) and in lines 95-96 they exclude studies focused on bacteria, fungi and viruses.

Authors: The word “parasitic” has been removed.

  1. Line 86: “parasites” AND primate genus.

Authors: Done.

  1. line 101: instead of the parasite, please, consider the use of the nature of the sample, i.e., skin and/or fur.

Authors: The type of biological simple is specified in Table 2.

  1. Line 107: replace “take into account” with another more formal verb, e.g. consider.

Authors: Done.

  1. Line 123: Table 1.

Authors: Done.

  1. Lines 137-141: if the authors are talking about Europe, they should rewrite these lines as follows: “Additionally, there were... Europe: Aotus,... (1 record each), Cebus in France (1 record), Saguinus in Italy (1 record),...” and then the species from China, Japan, Korea and South Korea.

Authors: It was rewritten as “Additionally, there were records for captive Neotropical primates in Europe and Asia: Aotus, Callimico, and Cebuella in Switzerland (1 record each), Cebus in France (1 record), Saguinus in Italy (1 record), Ateles, Saimiri, and Sapajus in China (1, 2, and 1 records, respectively), Callithrix in Korea (1 record), and Saimiri in Japan and South Korea (1 record for each country).”

  1. Line 148: endangered.

Authors: Done.

  1. Line 156: Alouatta ululata, OR Cebus cesarae.

Authors: Done.

  1. Line 161: data have been (“data” is assumed as plural noun).

Authors: Done.

  1. Line 173: to the least reported.

Authors: Done.

  1. Line 213: the authors say “only few studies” and give two references. If this is the case, they should say “only 2 studies” and be more precise.

Authors: Done.

  1. Line 223: “standardization of the result presentation/display”.

Authors: Done.

  1. Line 226: Remove the bracket and replace “In this way” with a more formal synonym. 244. Line 244: remove quotation marks.

Authors: Done.

Major comments.

In my opinion, the authors should considerably work on table 3 and the information has to be displayed more clearly.

  1. For the organisation of the taxa, I would suggest:
  2. Acanthocephala,
  3. Protozoa,
  4. Trematoda,
  5. Cestoda,
  6. Nematoda,
  7. Arthropods (different families).

Authors: Taxa were organized as suggested.

  1. Regarding the parasites’ names, there are so many repetitions, please look at them carefully and remove duplicates.E.g. Protozoa in Alouatta caraya (e.g. Giardia duodenalis is the same as G. lamblia).

Authors: Indeed, in Table 3 there are parasites with synonymous names, as parasites’ names were included exactly as reported in the publications. Aiming to clarify, we included this statement in the caption of Table 3 and in the Supplementary Material: “Parasites’ names were included exactly as reported in the retrieved publications.”

  1. Furthermore, to make the table shorter, when more than one specie of the same genus is mentioned, the next ones should be written with the abbreviation and the forward slashes must be removed:Plasmodium malariae, P. brasilianum, P. ovale.

Authors: We agree and we followed this rule in the text. In the table, we would prefer to keep the entire name of species when referred to different hosts, in order to facilitate the reading. The forward slashes were not removed, as parasites’ names were included exactly as reported in the retrieved publications (see legend to table 3).

  1. On many occasions, the authors specify the species and they give the abbreviation “sp.” before (e.g. See Parasites with zero prevalence in Aotus infulatus). Please remove sp. and give only the names of the species.

Authors: Parasites’ names were included exactly as reported in the publications. There are several cases of publications reporting both the name of the species and the genus plus sp. .Therefore, parasites with the abbreviation “sp.” have also been included when so reported.

  1. It would be better to make “groups” when naming the parasites and put all the species from the same genus together. E.g. Protozoa in Alouatta palliata. Put all Amoebas together. E.g. Protozoa in Alouatta seniculus. Put all Plasmodium together.

Authors: We grouped the parasites as suggested.

  1. The authors sometimes mention a parasite family and immediately they write species belonging to that family. Please, in these cases, writhe the name of the species in bracket or remove the family name. It is very confusing and it sometimes gives the impression that the authors do not have clear which genera belong to certain families of parasites.

Authors: We agree that this information may be confusing, however, parasites’ names (families, genera, species) were included exactly as reported in the publications, therefore, parasite family and species belonging to that family are both included without brackets. Aiming to clarify, we included this statement in the caption of Table 3 and in the Supplementary Material: “Parasites’ names were included exactly as reported in the retrieved publications.”

  1. As for the references, there should be a standardisation when they are cited in the table: [23], [50], or [23, 50]

Please, use a dash when consecutive references are used: [24-30], [51-53].

Authors: Done.

With regard to the discussion:

  1. Lines from 166 to 182 should be placed in the result section.

Authors: The following paragraph was placed into the results section:“Alouatta was the NHPs genus with most records (n=51) followed by Callithrix (n=22), while for all other genera less than 20 records were retrieved (Table 1). Considering the number of species, the genera with more species are Plecturocebus (n=23), Saguinus (n=15), Cebus (n=14), Alouatta (n=12), and Aotus (n=11). Likewise, Ateles, Brachyteles, Callimico, Callithrix, Cebuella, and Sapajus had records for 100% of the species, and 80% of the species for Lagothrix, Pithecia, and Saguinus. Brazil was the country with most parasitological records (n=163), being also the country with highest recorded occurrence of NHPs. Other NHPs rich-countries, such as Peru and Colombia, were the second and third countries with more records (20 and 13, respectively).”

  1. Line 198: there are so many references, here. Please, select the 5 most relevant.

Authors: We selected one reference per each parasite.

  1. The discussion is quite poor, and more relevant information should be added regarding the parasites found in Plathyrrhines and the whole process of review.

Authors: Thank you for the suggestion, we added more information into the discussion section (Lines 1187-1195, 1198-1203, 1213-1219, 1223-1231, 1243-1245).

Round 2

Reviewer 1 Report

As a result of the review the authors have considerably improved the manuscript. Now it can be accepted for publication.

Reviewer 3 Report

Dear authors and Editors,

In my opinion, the authors have addressed the comment very well with the given suggestions. Just a quick comment, please check the "/" in some parasite species and use the proper name, e.g. Plasmodium xxx, P. yyy.

Without any further ado,

Regards.